# Development of BCR-ABL1 Transgenic Zebrafish Model Reproducing Chronic Myeloid Leukemia (CML) Like-Disease and Providing a New Insight into CML Mechanisms

**DOI:** 10.3390/cells10020445

**Published:** 2021-02-19

**Authors:** Daniela Zizioli, Simona Bernardi, Marco Varinelli, Mirko Farina, Luca Mignani, Katia Bosio, Dario Finazzi, Eugenio Monti, Nicola Polverelli, Michele Malagola, Elisa Borsani, Giuseppe Borsani, Domenico Russo

**Affiliations:** 1Unit of Biotechnology, Department of Molecular and Translational Medicine, University of Brescia, 25123 Brescia, Italy; varinellimarco93@gmail.com (M.V.); l.mignani001@unibs.it (L.M.); dario.finazzi@unibs.it (D.F.); eugenio.monti@unibs.it (E.M.); 2Unit of Hematology, Department of Clinical and Experimental Sciences, University of Brescia, Bone Marrow Transplant Unit, ASST Spedali Civili, 25123 Brescia, Italy; simona.bernardi@unibs.it (S.B.); mirkfar@gmail.com (M.F.); katia.bosio49@gmail.com (K.B.); nicola.polverelli@unibs.it (N.P.); michele.malagola@unibs.it (M.M.); domenico.russo@unibs.it (D.R.); 3Centro di Ricerca Emato-Oncologica AIL (CREA), ASST Spedali Civili, 25123 Brescia, Italy; 4Laboratorio Centrale Analisi Chimico-Cliniche, ASST Spedali Civili, 25123 Brescia, Italy; 5Division of Anatomy and Physiopathology, Department of Clinical and Experimental Sciences, University of Brescia, 25123 Brescia, Italy; elisa.borsani@unibs.it; 6Unit of Biology and Genetic, Department of Molecular and Translational Medicine, University of Brescia, 25123 Brescia, Italy; giuseppe.borsani@unibs.it

**Keywords:** zebrafish, chronic myeloid leukemia (CML), BCR-ABL1, transgenic animal model, UAS/hsp70-Gal4, embryonic development, hematopoiesis

## Abstract

Zebrafish has proven to be a versatile and reliable experimental in vivo tool to study human hematopoiesis and model hematological malignancies. Transgenic technologies enable the generation of specific leukemia types by the expression of human oncogenes under specific promoters. Using this technology, a variety of myeloid and lymphoid malignancies zebrafish models have been described. Chronic myeloid leukemia (CML) is a clonal myeloproliferative neoplasia characterized by the *BCR-ABL1* fusion gene, derived from the t (9;22) translocation causing the Philadelphia Chromosome (Ph). The BCR-ABL1 protein is a constitutively activated tyrosine kinas inducing the leukemogenesis and resulting in an accumulation of immature leukemic cells into bone marrow and peripheral blood. To model Ph+ CML, a transgenic zebrafish line expressing the human *BCR-ABL1* was generated by the Gal4/UAS system, and then crossed with the hsp70-Gal4 transgenic line. The new line named (*BCR-ABL1*pUAS:CFP/hsp70-Gal4), presented altered expression of hematopoietic markers during embryonic development compared to controls and transgenic larvae showed proliferating hematopoietic cells in the caudal hematopoietic tissue (CHT). The present transgenic zebrafish would be a robust CML model and a high-throughput drug screening tool.

## 1. Introduction

Thanks to anatomical and genetic similarities with human and other vertebrates, Zebrafish (*Danio rerio*) is an in vivo experimental model providing exceptional advantages in studying physiological or pathological processes and pathogenesis of different human diseases [1,2]. The vast majority of hematopoietic malignancies arise through chromosomal alterations or insertions that lead to abnormal activation of oncogenes. Identifying which genetic lesions cause neoplasia and how mutations or chromosomal translocations may activate leukemogenesis is a primary objective in oncologic research. During the latest years, zebrafish model was increasingly used to study human malignant hematopoiesis by generation of transgenic lines expressing human mutant genes or carrying genomic alterations involved in leukemogenesis [3]. Zebrafish hematopoiesis shares, with humans, all lineage blood cells which are generated by similar developmental pathways and regulated by the same genes and transcriptional factors [4]. Moreover, the genetic and experimental accessibility of the fish and the optical transparency of its embryos and larvae make it an optimal tool for in vivo analysis of hematopoiesis [3,5]. Indeed, the recent development of powerful genome editing approaches, genome-wide forward genetic screens and chemical screening tools generated zebrafish models that recapitulate human malignant hematopoietic pathologies and unrevealed cellular mechanisms involved in these diseases [6,7]. Transgenic technology enables the generation of specific types of tumors by the expression of human oncogenes under tissue specific promoters. The vast majority of zebrafish leukemia models relies on overexpression of proto-oncogenes that are deregulated in different human hematologic malignancies, such as acute myeloid leukemia [6,7], acute lymphoblastic leukemia [8,9]; chronic lymphocytic leukemia [10]; and lymphomas [9,11]. The described pathologies are characterized by relevant multiple genetic mutations, but in many cases they are not good models because rarely the pathogenic mechanism and progression of neoplastic disease is due to a single driver mutation [12,13]. Conversely, Philadelphia chromosome (Ph) is the single driver alteration of Ph positive chronic myeloid leukemia (CML). CML is a clonal myeloproliferative disease [14] characterized by the breakpoint cluster region proto oncogene 1 tyrosine protein kinase (BCR-ABL1) fusion gene, derived from the t(9;22) translocation [15,16,17]. The BCR-ABL1 protein (P210) is a constitutively activated tyrosine kinase, causing anomalous activation of intracellular signal transduction pathways [18,19]. Many downstream genes result deregulated, including those involved in JAK/STAT [20] and RAS signaling [21]; FOXO pathway [22,23]; cyclins and chaperone [24]. These lead to genomic instability [25], abnormal cellular proliferation and cell survival [24], and finally to selection and amplification of more aggressive CML sub-clones [26], with an accumulation of immature hematopoietic stem cells into the bone marrow and the peripheral blood [27]. This alteration clinically translates in the progression of disease from chronic to blastic phase. Several mouse models miming BCR-ABL1 driven leukemogenesis (32,33) have been established [28] both to characterize the disease and to test different therapeutic compounds, particularly the BCR-ABL1 tyrosine-kinase inhibitors (TKIs) [29,30,31,32]. Many efforts led to the availability of murine CML models suitable for in vivo test of specific TKIs [33,34], and allowing the investigation of key players in disease development [35,36,37]. *Drosophila melanogaster* has been used as an alternative CML animal model [38,39] but, at present, only one stable zebrafish model for a CML-like disease has been recently generated [40]. Xu and colleagues recently reported the generation of a transgenic zebrafish line using a synthetic human *BCR-ABL1* transcript introduced by a transgenic construct in zebrafish embryos. After multiple activation of *BCR-ABL1* expression, between six months and one year the fish developed a myeloproliferative disease resembling human CML, as expected considering the median human CML onset [41,42,43]. In the present paper, we are going to introduce a new CML zebrafish model developed using a human BCR-ABL1 transcript isolated from a CML patient. The human fusion transcript under the control of the Gal4/upstream activating sequence (UAS) has been used to generate a transgenic line using Tol2 transposition system and then crossed with the *hsp70* (heat inducible promoter) Gal 4 line to recapitulate hematologic characteristics and molecular biology features of CML. The presenting model is featured by an early development of CML-like disease in embryos and is proposed as a new tool for a real-time observation of leukemic cells pathogenesis and analysis of signaling pathways affected downstream the fusion gene.

## 2. Materials and Methods

### 2.1. Bioinformatic Analysis of bcr and abl1

Genomic sequences of zebrafish *bcr* and *abl1* genes were analyzed using the Ensembl Genome Browser GRCz11 and ZFIN Database. For *bcr* in silico analysis we used the following accession numbers (ENSDART0000000191864) and (ZFIN ZDB-GENE-040724-43) and for Bcr protein accession number (ENSDARP00000147366) while for gene sequence of *abl1* (ENSDART00000148626) and (ZFIN ZBD-GENE-100812) and Abl1 protein (ENSDARP00000124744). Nucleotide and amino acid sequences were compared to the non-redundant sequences present at the NCBI (National Center for Biotechnology Information) using the Basic Local Alignment Search Tool (BLAST). Multiple sequences alignment was performed using the MUSCLE algorithm; accession numbers Bcr (XP_002665983) and Abl1 (XP_005172104). Human Proteins Accession number BCR (NP_004318) and ABL1 (NP_005148).

### 2.2. Animals Husbandry

Animals were staged, fed and mated as described by Kimmel et al. [44] and maintained in a largescale aquaria system (TECNIPLAST-BUGUGGIATE VARESE). Embryos were obtained by natural mating, raised at 28 °C in Petri dishes containing fish water (50×: 25 gr Istant Ocean; 39.25 gr CaSO4 and 5 gr NaHCO3 for 1 L) and kept in 14–10 h light-dark cycle. All experimental procedures complied with European Legislation for the Protection of Animals used for Scientific Purposes (Directive 2010/63/EU). All animal experiments were performed in accordance with the standards defined by Local Committee for Animal Health (Organismo per il Benessere Animale) and the permission of Italian Ministry of Health (287/2018-PR).

### 2.3. Microinjection

Injections were carried out on one-cell-stage embryos (with Eppendorf FemtoJet Micromanipulator 5171); the dye tracer phenol-red was co-injected as a control. After microinjection, embryos were incubated in fish water at 28 °C. Embryo development was evaluated at 24 h post fertilization (hpf), 48 hpf, and 72 hpf.

### 2.4. Generation of the pToL BCR-ABL1 pUAS-CFP/Hsp70-gal4 Transgenic Line

A transposon-donor plasmid named pTol2, containing the *BCR-ABL1* human cDNA linked with Cyan Fluorescent protein (CFP) and under the control of UAS promotor was used. We co-injected 500 pg of the plasmid pUAS:CFP/*BCR-ABL1*-Tol2 (linearized with Sal1) with 250 pg of in vitro synthesized Tol2 transposase mRNA [45,46] into wild type (AB strain) embryos at one-cell stage [47]. Microinjected embryos were activated at 40 °C at 15 somites (see Section 2.5), selected at 24 hpf for their mosaic transgenic expression using an epifluorescent microscope and were allowed to grow till 3 months. Out of 250 embryos injected with the recombinant construct (p*UAS:CFP/BCR-ABL1*-Tol2), about 100 representing F0 founder fish mosaic for expression of the transgene. At 3 months tail fin was excised from each individual fish for genotyping, DNA was extracted and amplified using specific primers (see Table 1). PCR-amplified DNA was subjected to analysis to evaluate whether the fish carries the transgene and agarose gel electrophoresis was used for the detection of positive samples. Fish positive for the transgene were then out-crossed with wild type AB fish to produce the Tg(UAS:CFP-*BCR-ABL1*) ±, subsequently crossed with Tg(hsp70:GAL4) line to produce the pure transgenic line tg *BCR-ABL1*pUAS/hsp70-Gal4 as schematically illustrated in Appendix A.

### 2.5. Activation of the Transgenic Promoter hsp70

Tg *BCR-ABL1*pUAS/hsp70-Gal4 embryos at 15–20 somites stage were transferred in a Petri dish with prewarmed fish water at 40 °C. The embryos were incubated for 1 h at 40 °C, then placed in fish water at 28 °C. The activated embryos were collected at different stages (24, 30, 36, 48 and 72 hpf) for different experiments. The fluorescence appeared after 2 h of activation by heat-shock and increased until 12 h post activation [48]. Gal4 expressing fishes were grown till adult age and then mated to collect embryos for experimental analysis.

### 2.6. Whole-Mount in Situ Hybridization (WISH) Analysis

Whole-mount in situ hybridization was performed following the procedure described by Thisse and colleagues [49]. Briefly, embryos were collected, dechorionated and incubated at 28 °C at different stages. Embryos were fixed overnight in 4% paraformaldehyde (PFA) at 4 °C, dehydrated through an ascending methanol series and stored at −20 °C. After treatment with proteinase K (10 μg/mL, Roche), the embryos were hybridized overnight at 68 °C with DIG-labelled antisense RNA probes. Embryos were washed with ascending scale of Hybe Wash/PBS and SSC/PBS, then incubated with anti-DIG antibody conjugated with alkaline phosphatase over night at 4 °C. The staining was performed with NBT/BCIP (Roche) substrates. All incubations of wild type and tg BCR-ABL1 fish were carried out together, the wild type embryos had the tip of the tail cut for their *post hoc* recognition. The described protocol was used for the following probes: *lmo2; c-myb; gata1; pu.1; mpx; scl. WISH* images were taken using a Zeiss Axio Zoom V16 equipped with Zeiss Axiocam 506 color digital camera and processed using Zen Pro Software from Zeiss.

### 2.7. RNA Extraction and Retrotrascription

For expression analysis, RNA was extracted from pools of 30 embryos at 24 hpf using TRI-Reagent (Sigma-Aldrich Milano, Italy) according to manufacturer’s protocol. RNA was quantified using the My Spect spectrophotometer (Biomed-Bari, Italy) and quality controlled by electrophoretic separation on a 1% TAE-agarose gel. 1.0 μg of total RNA was retro-transcribed to cDNA using Im-Prom Reverse Transcriptase (Promega) and oligo(dT) primers following the manufacturer’s protocol.

### 2.8. Digital PCR BCR-ABL1 Transcript Quantification

*BCR-ABL1* transcript absolute quantification was performed by digital PCR (dPCR), due to its reported sensitivity in human *BCR-ABL1* transcript detection [50]. dPCR assay was performed by the chip-based dPCR platform system Quant Studio 3D (QS3D) (Thermo Fisher Scientific), as previously described [51,52,53]. Briefly, a reaction mix containing 8 μL of 2X QuantStudio 3D Digital PCR Master Mix (Thermofisher Scientific), 0.8 μL of 20X TaqMan-MGB-FAM-probe customized assay, 5 μL of cDNA, and 2.2 μL of nuclease-free water was prepared. The negative control reaction mix contained 8 µL of 2X QuantStudio 3D Digital PCR Master Mix, 0.8 μL of 20X TaqMan-MGB-FAM-probe assay, and of 7.2 μL nuclease-free water. For each sample, we loaded 15 μL of the reaction mix onto a QuantStudio 3D Digital PCR 20K Chip (Thermofisher Scientific) by the automatic chip loader. The target was amplified using the following thermocycling profile: 95 °C for 8′, 45 cycles at 95 °C for 15”, and 60 °C for 1′, with a final extension step at 60 °C for 2′ on ProFlex PCR system. The chips were then imaged by dPCR QS3D and a secondary analysis was performed by QuantStudio 3D AnalysisSuite Cloud Software. Positivity emission threshold was fixed at 4000 RFU, as previously reported [54].

### 2.9. Real Time PCR Transcripts Quantification

Real-Time PCR (RT-PCR) was performed using the Eco-Illumina system. Reactions were performed using a final volume of 20 μL, with 2 μM of primer, 12.5 μL of Syber Green Master Mix (Biorad-Italy) and 20 ng of cDNA. The cycling parameters were: 95 °C for 1 min, followed by 40 cycles of 95 °C for 15 s, 60 °C for 35 and 72 °C for 25 s). Each reaction was performed in triplicate and alfa-elongation factor 1α was used as internal standard in each sample. Threshold cycles (Ct) and melting curves were generated automatically by Rotor-Gene Q series software. Relative quantification was calculated by the ddCT method as described previously [55]. Sequences of specific primers used in this paper for RT-PCR are listed in Table 1. Primers were designed by the PrimerQuest and Real Time PCR Tool by IDT software.

### 2.10. Acridine Orange Staining

To analyze the level of cells death, acridine orange staining was performed using a standard protocol [56]. Embryos at 48 hpf were dechorionated and incubated for 30 min in acridine orange solution (10 mg/mL). Embryos were rinsed three times using PBS, mounted using 80% glycerol and quickly imaged using epifluorescent microscopy (Zeiss Axio Zoom V16 equipped with Zeiss Axiocam 506 colour digital camera and processed using Zen Pro software from Zeiss).

### 2.11. Cell Proliferation Assays H3 Immunostaining and BrdU Immunostaining

Embryos at 24 hpf were fixed overnight in 4% paraformaldehyde (PFA) at 4 °C, dehydrated through an ascending methanol series and stored at −20 °C. The first day the embryos were rehydrated in PBT and washed in Tris Buffer (150 mM Tris HCl pH 9.0) for 5 min. The embryos were then incubated in Tris Buffer 15 min at 70 °C and washed 2 times for 5 min in PBT and in H_2_O in ice. The embryos were then incubated in acetone at −20 °C for 20 min and treated in blocking buffer for 4 h at 4 °C. The embryos were incubated with the primary Ab PH3 in blocking buffer at 4 °C over-night, then washed 4 times for 30 min with PBT and incubated with the secondary Ab for 3 h at R.T in the dark. The embryos were washed 5 times for 5 min in PBD and fixed 20 min with PFA 4%, then washed 5 times 5 min in PBT and observed under the microscope [57]. Wild type and Tg BCR-ABL1 embryos at 24 hpf were incubated in 10 mM BrdU for 2 h. The embryos were then stained with mouse-anti-BrdU antibody followed by incubation with Alexa Fluor-488-anti mouse antibody for fluorescent visualization.

### 2.12. Blood Extraction

The fish transferred into 50 mg/L tricaine for 1–3 min. The blood was harvested from zebrafish by making a lateral incision just posterior to the dorsal fin in the dorsal aorta area and used to prepare blood smears. Slides were then stained with May-Grunwald-Giemsa stain and examined by microscopy [58].

### 2.13. Imaging

Images of CFP expression and bright field were taken with Axiovert 200 M Apothome 2 and processed using Zen Pro software from Zeiss. Whole mount in situ hybridization imaging and bright field imaging of transgenic and wild type embryos (anaesthetized with 0.04% tricaine, embedded in 0.8% low melting agarose and mounted on a depression slide) were captured using a Zeiss Axio Zoom V16 equipped with Zeiss Axiocam 506 color digital camera and processed using Zen Pro software from Zeiss.

### 2.14. Histomorphology Analysis

Immediately after suppression, animals were fixed in Bouin’s solution or 24 h, embedded in paraffin according to standard procedures and longitudinally cut at 8 μm by a microtome (Microm HM 325). Histomorphological assessment was made with the Hematoxylin and Eosin staining (HE; Bio-Optica, Milan, Italy), according to the manufacturer’s protocol. The images were acquired using an optical microscope (Olympus, Hamburg, Germany).

### 2.15. Quantification of Fluorescence Intensity by ZF-Mapper Application

Pictures of single embryos were taken with the same magnification (20X), laser intensity and exposition using Zeiss Axio Zoom V16 equipped with Zeiss Axiocam 503 color digital camera in 8 bit using Zen Pro software and exported in tiff format. Florescence was measured using Z-Mapper software [59] with a threshold value of 7; data are represented as the ratio between total pixel intensity and the number of pixels.

### 2.16. Statistical Analysis

All the experiments described in the manuscript were performed at least three times and we used GraphPad Prism software 6 to perform statistical analysis. The comparison and significance between different analyzed groups was determined by two-way ANOVA, corrected for multiple comparison or by two-tailed unpaired Student’s *t*-test The *p*-value are indicated with asterisks * *p* < 0.05%, ** *p* < 0.01%, *** *p* < 0.001%, and **** *p* < 0.0001%. Differences were considered significant at *p* values of less than 0.05.

## 3. Results

### 3.1. Generation of BCR-ABL1 Transgenic Line

The multiple alignments between human BCR and ABL1 and zebrafish orthologous protein sequences showed a high degree of identity. In particular, using the Clustal W Omega tool [60], we observed that zebrafish Bcr and Abl1 shared 76% and 72% identity with their human counterparts, respectively. Moreover, a deep in silico evaluation of zebrafish Abl1 protein with the human ABL1 counterpart indicated the presence of highly conserved tyrosine kinase domain and in particular the amino acids that are involved in bonds with tyrosine kinase inhibitors are preserved as reported in Appendix A. To investigate the oncogenic properties of BCR-ABL1, the human fusion oncogene was injected as cDNA in zebrafish embryos at one-cell stage, under control of the UAS promoter. The construct was designed to integrate the complete coding sequence into the host genome using the Tol2 transposition system, allowing the generation of transgenic fishes (Figure 1 Panel A, A). F0 founders with the highest germ line transmission rate were identified on the basis of caudal fin genotype (Figure 1 Panel A, B) and DNA sequence analysis (data not shown). Stable F1 generations were obtained by intercrossing the founder fishes with wild type fishes (AB line). At the mature age, the F1 generation was mated with a transgenic line carrying the inducible system hsp70-Gal4 and the expression of CFP protein was evaluated at 22 and 24 hpf after heat shock treatment. The stable transgenic line has been named Tg (*BCR-ABL1*pUAS:CFP/Hsp70-Gal4), here in after Tg BCR-ABL1. Intermediate cell mass (ICM) is a region of developing blood cells arising late in the segmentation period and located in the anterior-ventral tail and just posterior to the yolk extension. ICM is imaged in Figure 1 Panel A (C). As shown in Figure 1 Panel A (D–E), in the region corresponding to ICM, several of CFP positive cells are present at 22 and 24 hpf. The BCR-ABL1 transgene expression was confirmed by dPCR (Figure 1 Panel B).

### 3.2. Induced Expression of BCR/ABL1 Increased the Number of Myeloid Cells during Embryonic Development

To explore the effects of *BCR-ABL1* oncogene expression in zebrafish, we crossed the transgenic line and induced transgenic expression by heat-shock activation. We then observed and analyzed embryos for hematopoietic phenotype during embryonic development. At 24 hpf, wild type embryos do not show any abnormality. By contrast, the Tg BCR-ABL1 embryos showed an accumulation and proliferation of hematopoietic cells in the caudal region (Figure 2 Panel A). The blood accumulation site in Tg BCR-ABL1 embryos corresponds to ICM, in this region we observed the expression of CFP positive cells (Figure 1 panel A,C–E), that contains multipotent myeloid progenitor cells (MPCs) capable of producing cells of erythroid and other myeloid lineages. We used phosphohistone 3 (PH3) immunostaining to label cells in M phase and quantify them (Figure 2 Panel A,E): in wild type embryos 13.6% (±1.7) of cells were PH3 positive while in Tg BCR-ABL1 45.6% (±1.22) of cells were labelled by PH3 staining (*p* < 0.0001). At 48 hpf, when the Hematopoietic Stem Cells (HSCs) enter the circulation and migrate to colonize the Caudal Hematopoietic Tissue (CHT) in the posterior tail region, we still observed proliferation in Tg BCR-ABL1 fishes compared to wild types (Figure 2, Panel B). In order to assess the proliferation rate in CHT, we then used BrdU to label proliferating cells at different developmental stages (24, 48 and 72 hpf). Particularly, at 48 hpf in Tg BCR-ABL1 we observed a higher number of proliferating cells (69% ± 1.47) compared to wild type fishes (12.25% ± 0.47) (*p* < 0.0001) (Figure 2, Panel B); similar results were obtained at 24 and 48 hpf. In addition to abnormal cells proliferation, we investigated the resistance to apoptosis, another main feature of CML cells. At 24 and 48 hpf, we stained with acridine the Tg BCR-ABL1 embryos showed a lower level of apoptosis than wild type (Appendix A); the obtained results were supported by the quantification of the incorporated fluorescence at 24 and 48 hpf as reported in Figure 2 panel B (D). The data showed that transgenic fishes displayed abnormal myeloid cell expansion as result of increased cell proliferation on one hand and of reduced apoptosis on the other.

### 3.3. BCR-ABL Deregulates Lineage-Specific Hematopoietic Transcription Factors

In order to evaluate the impact of BCR-ABL1 expression during early phase of hematopoiesis, first we quantified by Real Time-PCR the transcripts level of the key players in the regulation of normal hematopoiesis. The *scl*, *gata1, lmo2* and *c-myb* genes are all expressed in primitive erythroid cells, *pu1* is one of the master regulator involves in normal myeloid cell/B-lymphocytes and *mpx*, the granulocyte specific gene which encodes for an enzyme abundant in the granules of mature neutrophils [61]. At 24 hpf, in tg BCR-ABL1 embryos *lmo2*, *gata1* and *scl* resulted upregulated compared to wild-type counterpart. As expected, *pu1* expression level was highly upregulated as well as *mpx* that resulted consistently increased with the increased expression of *pu1* in Tg BCR-ABL1 when compared to wild type embryos. Finally, the expression level of *c-myb* in Tg BCR-ABL1 fishes did not decrease with cell growth and differentiation when compared with wild type (Figure 3).

By the end, we observed that BCR-ABL1 induced granulopoiesis expansion and granulocytes differentiation (Figure 4). CML is distinguished by clonal expansion of primitive pluripotent stem cells without the loss of their capability to differentiate into myeloid cells and by a dynamic increase in proliferation of granulocytic cells. Based on the results obtained by RT-PCR and the high increase of *pu1* expression level, we wondered whether the increased *pu1* expression observed in Tg BCR-ABL1 fish may affect granulocytic and/or monocytic cell fates. Using WISH technique at 24 hpf, we firstly analyzed the expression of *scl*, one of primitive transcription factor involved in hematopoietic differentiation. The staining for *scl* was clearly increased in Tg BCR-ABL1 compared to wild type fish (Figure 4A,B). Similarly, we followed the embryonic development at 28 hpf and we found a more intense staining for *pu1* in the vast majority (90% *n* = 68/75) of Tg BCR-ABL1 embryos when compared with wild type ones (Figure 4C,D). Successively, we looked for the presence of *mpx* positive granulocytes and macrophages, expressing *L-plastin*. *L-plastin* is a pan-myeloid marker that identifies all myeloid subsets and it is predominantly expressed in macrophages and monocytes. As shown in Figure 4, while wild type embryos showed a low number of *mpx* and *l-plastin* positive cells (violets dots) in the trunk region corresponding to ICM (Figure 4E–G), the Tg BCR-ABL1 fish showed a dramatic expansion of *mpx* positive cells in a large number of embryos (93% *n* = 70/75) in ICM and in the anterior part of the yolk. Similar result was obtained by *l-plastin* hybridization. (Figure 4F–H). We performed a quantification of positive cells in ICM and observed a significant difference for *mpx* (21.2 ± 0.25 vs 8.6 ± 0.25 per embryo, *p* < 0.0001) and *l-plastin* (34.8 ± 1.4 vs 12.2 ± 0.66 per embryo, *p* < 0.0001) in Tg BCR-ABL1 compared to wild type embryos, respectively (Figure 4I,L). Overall, these results suggest that BCR-ABL1 reprograms the cell fate decision of the multipotent progenitor cells towards the myeloid lineage.

### 3.4. BCR-ABL1 Down-Regulates Erythropoiesis and Neovascularization

Based on the above described results, we thought to understand the behavior and specification of erythroid lineage in Tg BCR-ABL1. To this purpose, we performed WISH experiments using different erythroid markers: *gata1*, *c-myb* genes which are expressed in primitive erythroid cells and required in ICM definitive hematopoiesis and *VE-cadherin* (*cadh5*) as a vascular endothelial-specific expression marker. We evaluated the expression level at 28 hpf and observed that *c-myb* did not decrease so strongly with cell growth and differentiation in Tg BCR-ABL1 embryos (*n* = 75/78) compared to wild type ones (Figure 5A,B). Conversely, a strong reduction of *gata1* mRNA level was observed in the vast majority (93% *n* = 73/78) of Tg BCR-ABL1 embryos compared to wild types (Figure 5C,D). In addition, at 30 hpf, a strong down-regulation of *VE-cadherin* mRNA was highlighted in fish expressing BCR-ABL1 (*n* = 69/78), resulting in an abnormal vascularization (particularly in caudal plexus) and a presence of shorter nascent vessels compare to wild type fish (Figure 5E,F). Altogether these results suggest that BCR-ABL1 promotes primitive myelopoiesis by inhibiting specific genes involved in erythroid lineage commitment.

### 3.5. Peripheral Blood Analysis in Tg BCR-ABL1

Patients with CML typically present an unregulated growth of myeloid cells, leading to a higher prevalence of myeloid precursor, basophil, eosinophils and macrophages cells in the bone marrow (BM) and in the peripheral blood (PB). To gain insight into the main characteristics of CML-like disorder, we collected peripheral blood from wild type and 15 months old fish expressing human BCR-ABL1. We analyzed the peripheral smear and performed blood cells count on a total of 15 animals. Erythrocytes were predominant and easily recognized by eosinophilic and ellipsoidal nuclei oriented parallel to the cell’ long axis. Small lymphocytes and thrombocytes had a similar appearance by light microscopy, lymphocytes are slightly larger than thrombocytes and, cytoplasm was much more evident. Cytoplasmic granules were prominent in granulocytes, and nuclear chromatin was dispersed throughout the nucleus. Eosinophil-like cells demonstrated specific granules with a rod- or disc-shaped crystalline structure similar to previous descriptions of peripheral blood cells from other fish models [1,61]. The blood cells from wild type fish were predominantly erythrocytes, with myeloid cells only occasionally observed (1.7%) as well the other cell types (Figure 6A) (Table 2 and Figure 7). By contrast, erythrocytes were significantly reduced to 28.4% in Tg BCR-ABL1 (Figure 6B and Table 2) and peripheral blood was extensively enriched by abundant blast-like cells (15.2% vs 1.5). They had cytoplasm larger than erythrocytes and higher nuclear to cytoplasmic ratio, containing multiple large nucleoli (Figure 6D,F). We also observed some phenotypes accompanying these CML-like tg BCR-ABL1 adult zebrafish including the presence and increases of macrophages (23.9% vs 2.5%), neutrophilic and eosinophilic granulocytes and lymphocytes (10.4% vs 1.7%) (Figure 6C–F and Table 2 and Figure 7). Thrombocytosis is present in approximately half of all newly diagnosed CML patients [62], interestingly 7 out of 15 (45%) of Tg BCR-ABL1 fishes presented thrombocytosis with counted platelets around 1.7% compared to 0.13% in wild type embryos, as reported in Table 2 and shown in Figure 6G. Taken together, the reported increased number of blasts aggregate cells with the proliferation of mature granulocytes (neutrophils and eosinophils), large number of macrophages, and thrombocytosis resemble some of the clinical features of CML.

### 3.6. Histologic and Phenotypic Analysis of Hematopoietic Organs in Tg BCR-ABL1

In adult zebrafish kidney is a primary hematopoietic organ where all blood cell lineages and their precursor are found; spleen contains mainly erythrocytes and thrombocytes (red pulp). The ellipsoids contains macrophages and reticular cells [63,64]. To study the tissue phenotype of the major hematopoietic organs in adult zebrafish (15 months old) we performed histological analysis in five Tg BCR-ABL1 and wild type fish. In spleen, we observed abnormalities and disorganization in ellipsoid structures of fish expressing human BCR-ABL1 compare to wild type (Figure 8A,B). In kidney, the hematopoietic tissue as well the collecting ducts resulted well-defined in wild type fish. On the contrary, in Tg BCR-ABL1 we observed few big and undefined cells, mirroring blasts aggregates, and a massive infiltration of blasts or immature cells closed to the collecting ducts (Figure 8C,D). The data suggest possible defects during hematopoiesis in Tg fish that resemble human CML.

## 4. Discussion

Zebrafish (*Danio rerio*) is a freshwater teleost fish that has emerged as in vivo model in many research fields. In the last decade, a significant number of human hematopoietic malignancies miming the crucial pathophysiological mechanisms of human counterpart disease have been modelled in zebrafish [65,66]. Recently, Xu and colleagues described a zebrafish model resembling a CML-like disease [40], useful model for studying Ph driven leukemogenesis. While considering the remarkable results, the model proposed by the Authors required multiple activations of *BCR-ABL1* expression by heat shock, thus it cannot be excluded that this method has forced the phenotype of the disease. Since *BCR-ABL1* is constitutively expressed and active in human CML, we generated a zebrafish CML model by using a GAL4/upstream activation sequence (UAS) transgenic system. This transgenic system has been widely used to regulate gene expression in a cell-specific and temporally restricted manner and provided a powerful tool to trace transgene expression during embryonic development, monitor subcellular structures, and target tissues for selective ablation or physiological analyses [48]. In order to characterize the BCR-ABL1 expressing model, we deeply investigated zebrafish hematopoiesis and also tracked the molecular changes preceding the morphological phenotype. Two main steps are described as primitive and definitive hematopoiesis, which take place in two different locations. The zebrafish primitive hematopoiesis takes place in anterior lateral mesoderm and ICM, a tissue derived from ventral mesoderm containing multipotent stem cells that give rise to the erythroid and myeloid cell lineages. A transient “intermediate” wave occurs in the posterior blood islands, where both erythrocytes and neutrophils are generated. Definitive hematopoiesis has been shown to initiate around 28–30 hpf and gives rise to definite adult-like HSCs, which have both self-renewal capacity and erythroid, myeloid and lymphoid potential. From 48 to 96 hpf, the newly emerged HSCs enter the circulation and migrate to colonize the CHT in the posterior zebrafish tail region for producing different cell lineages [3,67]. It is known that the oncogene *BCR-ABL1* is directly involved and has a causative role in CML [27] by promoting cell proliferation and inhibiting apoptosis [68]. Considering these pathogenic mechanisms, we firstly assessed the cellular proliferating rate by immunostaining for PH3 and BrdU. Both assays showed an increased number of proliferating cells in Tg BCR-ABL1 fish compared to wild type ones at different embryos developmental stages and confirmed the down-regulation of physiological apoptosis. These data proved the presence of the above mentioned key characteristics of human CML [69,70]. In addition to proliferative and apoptotic markers deregulation, human BCR-ABL1 is also known to hamper the physiological myeloid commitment. As described, in zebrafish there is a high conservation of transcription factors and major signaling pathways involved in HSCs differentiation and maturation [5]. We thought to investigate how BCR-ABL1 exerts the observed hematopoietic phenotype and to this purpose we performed RT-PCR gene expression studies in wild type and Tg BCR-ABL1 embryos at 24 hpf, an early phase of hematopoiesis process. We analyzed different myelo-erythroid gene expression markers: *c-myb*, *gata1*, *lmo2*, *mpx*, *pu1* and *scl*. Stem cells transcription factor *scl* is required in the promotion of primitive hematopoiesis [71] while *lmo2*, expressed in hematopoietic progenitors, acts in parallel with *scl* as an important hematopoietic regulator. Similarly, *gata1* is an early master regulator in erythrocyte development [72] as well as *pu1*, a transcription factor belonging to the Ets family, is a master gene involves in myelopoiesis and B lymphopoiesis. Finally, *c-myb* is predominantly present in immature hematopoietic cells and decreases during cell differentiation [73]. In addition, we considered *myeloperoxidase* (*mpx*), the granulocyte specific gene which encodes for an enzyme abundant in mature granules and it is considered the marker of mature neutrophils and eosinophils [74,75] and *L-plastin*, an actin-binding protein predominantly expressed in monocytes/macrophages. In our CML zebrafish model, the expression levels of these genes resulted unregulated and faulty. In particular, Tg BCR-ABL1 showed over-expression of *scl, lmo2*, *mpx*, and *pu1* and downregulation of *gata1*, meaning a hematopoiesis reprogramming by BCR-ABL1 towards the myelopoiesis and granulopoiesis and less towards the erythropoiesis. This alteration is well known and reported both in in vitro CML model [76] and in CML patients [77]. Focusing on CML patients, they typically develop a highly characteristic differential white blood cells count with high number of myelocytes-metamyelocites/monocytes-macrophages and segmented neutrophils. The CML natural progression is characterized by hyper-leukocytosis in peripheral blood and bone marrow with a prevalence of granulocytes and myeloid cells at different steps of maturation. The percentage of blast cells is less than 2% during the so-called CML chronic phase and progressively rise in the following accelerated and blastic phases [78,79,80]. To evaluate the presence of similar alterations in circulating blood cells, we investigated both the blood cells and the hematopoietic organs of our transgenic model. As reported in Figure 6, erythrocytes were significantly reduced in Tg BCR-ABL1, beside a high number of neutrophilic and eosinophilic granulocytes and macrophages. This peripheral blood difference has the feature of an altered myeloid differentiation typical of CML onset [81,82]. All the observed alteration resulted statistically significant when compared to the wild type fish. The evaluation of the primary hematopoietic organs also supported the CML phenotype activation driven by BCR-ABL1 expression. In adult zebrafish, kidney is known to be a primary hematopoietic organ and all blood cell lineages and their precursor are present, displaying a cellular complexity comparable to that of mammalian bone marrow. All cell lineages arise from kidney: mature erythroid cells, myeloid cells, (neutrophils, monocytes, macrophages and eosinophils), mature thrombocytes [83,84]. In addition, mature blood cells may be found in zebrafish spleen, mainly erythrocytes and thrombocytes (red pulp), while macrophages and reticular cells are pinpointed in the splenic ellipsoids [75]. In the adult Tg BCR-ABL1 fish, kidney and splenic tissue resulted altered and immature cells aggregates were observed (Figure 8) as also described in patients affected by CML [85,86] and other myeloid disorders [87]. Moreover, a level of 10–19% of blasts marks the transition from chronic phase to blast phase in human CML. Adults of the Tg BCR-ABL1 line (fish 22–24 months old) present an increase of blast cells in a rate of 10% in peripheral blood. Moreover, the animals are smaller in compare to wild types at the same age, swim slower and present skeletal alterations (data not shown).

Altogether, these exciting and remarkable evidences have an undeniable importance and strongly support the successful generation of a new CML zebrafish model, characterized by the expression of BCR-ABL1. This feature is of paramount relevance since it better resembles the natural physiopathology and leukemogenesis of human CML. Moreover, we demonstrate that molecular changes of key hematopoietic transcription factors and other different features are appreciable during embryonic development and in adult fish. Among them, of remarkable interest are the altered tissues of primary hematopoietic organs and CML-like blood cell count and differentiation. These results allow not only the monitoring of the pathological evolution, but also the application of high throughput test or drug screenings. In future studies, it will be interesting to go deeper inside molecular pathogenic mechanisms, to characterize the downstream genes that are deregulated by expression of the BCR-ABL1 fusion gene (JAK/STAT, RAS signaling; FOXO pathway; cyclins and chaperone) and to identify novel molecules with in vivo pharmacological effects.

In conclusion, our CML zebrafish model is a new valid tool to further investigate the molecular mechanism and biological characteristics of human CML disease and to easily and robustly test new therapeutic drugs for this paradigm of precision medicine, with still uncovered aspects.

## Figures and Tables

**Figure 1 cells-10-00445-f001:**
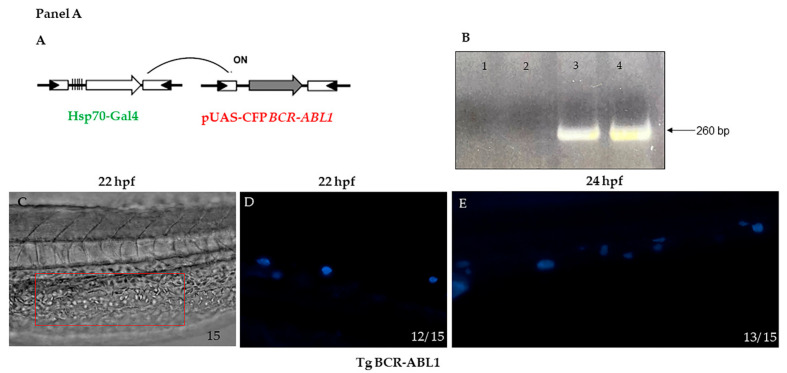
**Panel A**. (**A**) Schematic representation of strategy used to express the BCR-ABL1 fusion transcript from the Tg BCR-ABL1pUAS/hsp70-Gal4 fish. (**B**) Transgenic genotyping by PCR: tail fin from three months old fish was excised and cDNA amplified by specific primers; lane 1 negative control (water); lane 2 wild type fish; lane 3 transgenic fish expressing hBCR-ABL1; lane 4 positive control (plasmid) (**C**) Representative bright light image (phase contrast) of a trunk portion from 13 to 21 somites at 22 hpf corresponding to a region of intermediate cell mass (ICM) is boxed. (**D**,**E**) higher magnification of CFP expression in tg BCR-ABL1pUAS/hsp70-Gal4 after heat treatment; CFP is detectable in cells located in the region of PBI at early stages of development respectively at 22 hpf and 24 hpf. Lateral view and magnification 10 X C; lateral view and magnification 20 X D, E. Numbers in each panel represent total embryos observed for each experiment. Experiments were performed twice. **Panel B**. Digital PCR quantification of BCR-ABL1 transcript. BCR-ABL1 transcript was quantified by Quant Studio digital PCR system on Tg BCR-ABL1 and wild type embryos pools at 24, 48 and 72 hpf. Digital PCR graph represents the emission of fluorescence in the micro-reactions. Every pool was quantified twice on 2 different chips, each one divided in 20.000 micro reaction. Yellow dots represent negative micro-reaction (no emission), while blue dots represent positive reactions (emission in FAM). Each positive reaction contains one or two molecules of BCR-ABL1 transcript after Poisson distribution correction. hpf = hours post fertilization; WT = wild type.

**Figure 2 cells-10-00445-f002:**
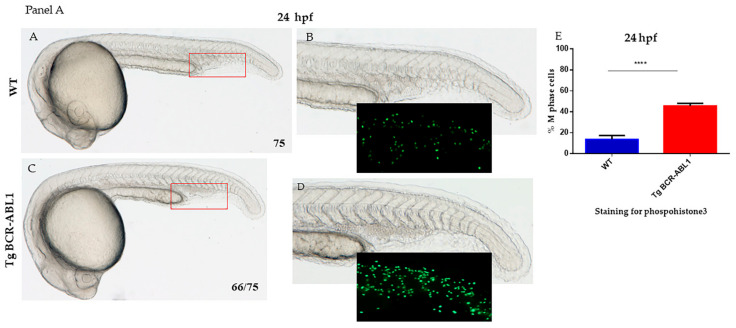
Panel (**A**). (**A**,**C**) Bright-light images of live wild type (WT) and Tg BCR-ABL1 at 24 hpf; the accumulated hematopoietic cells are indicated in the red box. (**B**,**D**) The areas in the red boxes are shown at higher magnification (12.5 X). The region (boxed) of posterior blood islands is enlarged and show the immunofluorescence staining of phospohistone3. All the experiments were performed at least in triplicate; images show one representative experiment. Numbers in each panel represent total embryos used for the experiment and embryos with the phenotype shown in the image. (**E**) Immunostaining with phospohistone 3 (PH3) to label cells in M-phase: percentage of PH3 positive cells over the total number cells present in posterior blood islands. (Magnification 7X A, C; 12.5 X B, D) Blue bar represents wild type; red bar represents Tg BCR-ABL1. Statistical analysis was performed by unpaired, two-tailed T Test **** *p* < 0.0001. Panel (**B**) (**A**,**B**) Bright light images of wild type (WT) and Tg BCR-ABL1 at 48 hpf. The images show a particular of the caudal hematopoietic tissue (CHT). Red arrows indicate cells accumulated in CHT. (**C**) The graph of BrdU staining show the percentage of proliferating cells in WT and Tg BCR-ABL1 at 24, 48 and 72 hpf; (**D**) Quantification of the fluorescence of embryos at 24 and 48 hpf shown in Appendix A acquired by ZF-Mapper software. Blue bar represents wild type; red bar represents Tg BCR-ABL1. All the experiments were done at least three times. Unpaired, two-tailed *t*-test *** *p* < 0.001, **** *p* < 0.0001.

**Figure 3 cells-10-00445-f003:**
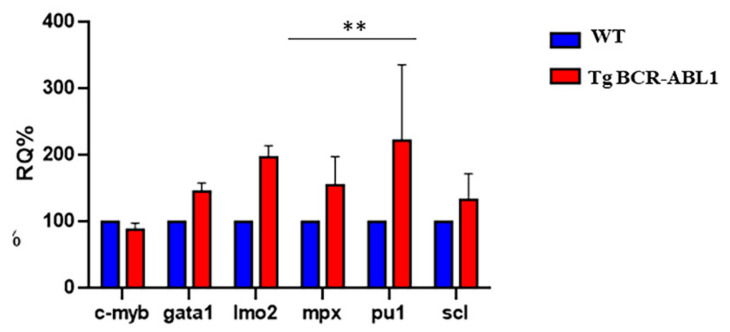
Expression level analysis of hematopoiesis markers. RT-PCR for transcript quantification of six known genes involved in hematopoiesis process. Gene expression was normalized using α elongation factor1 as a reference gene and expressed as relative quantification (RQ). Blue bars represent wild type; red bars represent Tg BCR-ABL1. Statistical analysis was performed by unpaired *t*-test; ** *p* < 0.01.

**Figure 4 cells-10-00445-f004:**
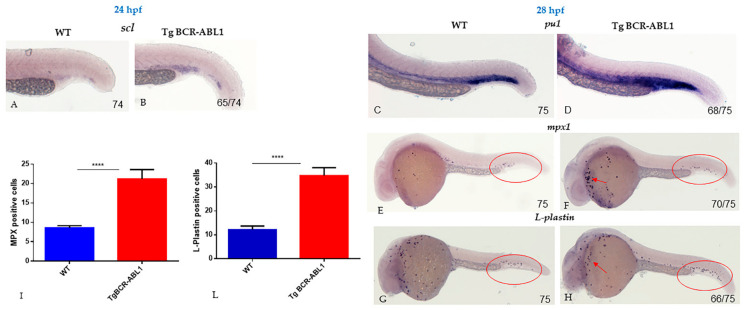
BCR-ABL1 induces a granulopoiesis expansion and granulocytes differentiation. Whole mount in situ hybridization (WISH) of scl at 24 hpf in Wild Type (WT) (**A**) and in Tg BCR-ABL1 (**B**). WISH at 28 hpf for *pu1* (**C**,**D**); *mpx1* (**E**,**F**); and *L-plastin* (**G**,**H**) respectively in WT and tg BCR-ABL1 fishes. The experiments were performed in duplicate; numbers in each panel represent total embryos used for the experiment and embryos with the result shown in the image. In each panel, red ovals indicate the positive signals of *mpx1* and *L-plastin* in posterior blood island. Red arrows respectively indicate the *mpx*1 (**F**) and *L-plastin* (**H**) positive cells in the anterior part of the embryos. Statistical analysis of *mpx1+* (**I**) and *L-plastin+* (**L**) signals counted in posterior blood islands. Unpaired, two-tailed *t*-test **** *p* < 0.0001.

**Figure 5 cells-10-00445-f005:**
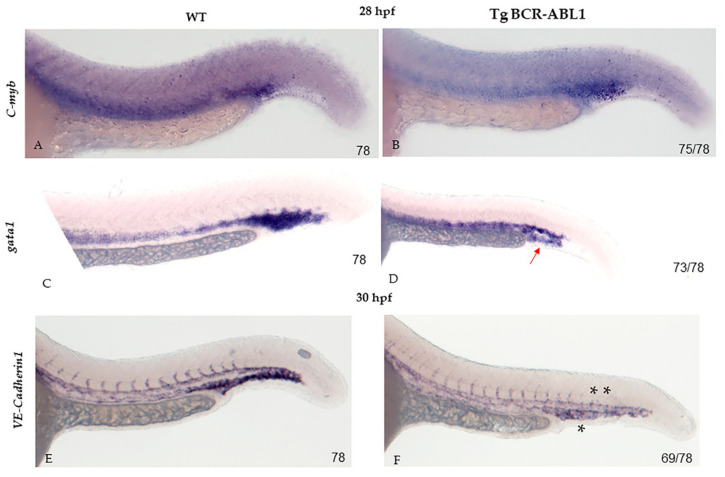
Transcriptional changes of erythropoiesis markers. Whole mount in situ hybridization (WISH) at 28 hpf of *c-myb* (**A**,**B**), *gata1* (**C**,**D**); *VE-cadherin* (**E**,**F**) in wild type (WT) and Tg BCR-ABL1 fish, respectively. The experiments were performed in duplicate; numbers in each panel represent total embryos used for the experiment and embryos with the result shown in the image. Red arrow in (**D**) indicate the downregulated expression level of gata1 in caudal plexus in Tg BCR-ABL1 fish; (**F**) single black asterisk indicate the downregulated expression level of VE-cadherin in caudal plexus in Tg BCR-ABL1 fish; double black asterisks indicate the not well-formed nascent vessels in Tg BCR-ABL1 fish.

**Figure 6 cells-10-00445-f006:**
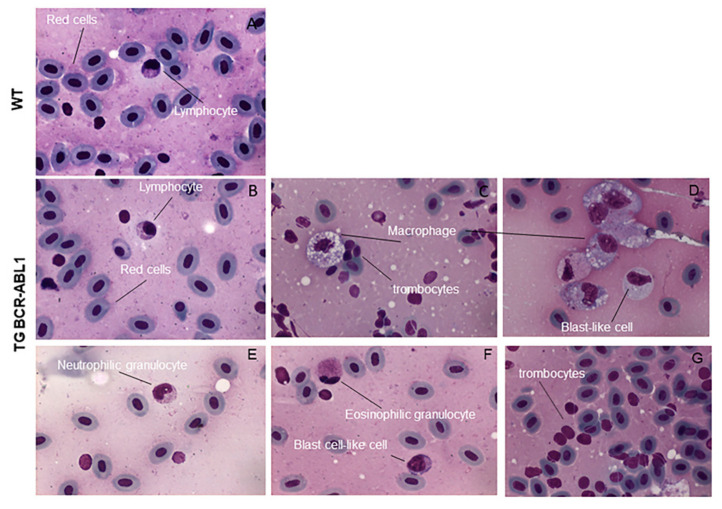
Cytological analysis of transgenic fish expressing hBCR-ABL1. May-Grunwald-Giemsa staining of peripheral blood cells from wild type (**A**) and Tg BCR-ABL1. Samples are representative of 15 wild type and Tg BCR-ABL1 fish 15 months old. Different types of cells are indicated in the pictures in (**C**–**G**).

**Figure 7 cells-10-00445-f007:**
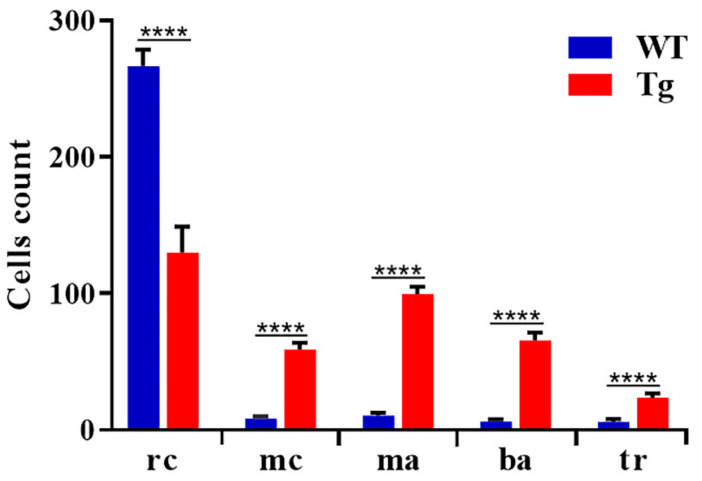
Quantification of different cell types in peripheral blood. Blood cells were counted manually based on their morphology and classified in different groups. We counted 500 cells per field for wild type (*n* = 15) and Tg BCR-ABL1 (*n* = 15). Statistical analysis was performed by unpaired, two-tailed T Test **** *p*< 0.0001. rc = red cells; mc = myeloid cells; ma = macrophages; ba = blast aggregates; tr = tromobocytes.

**Figure 8 cells-10-00445-f008:**
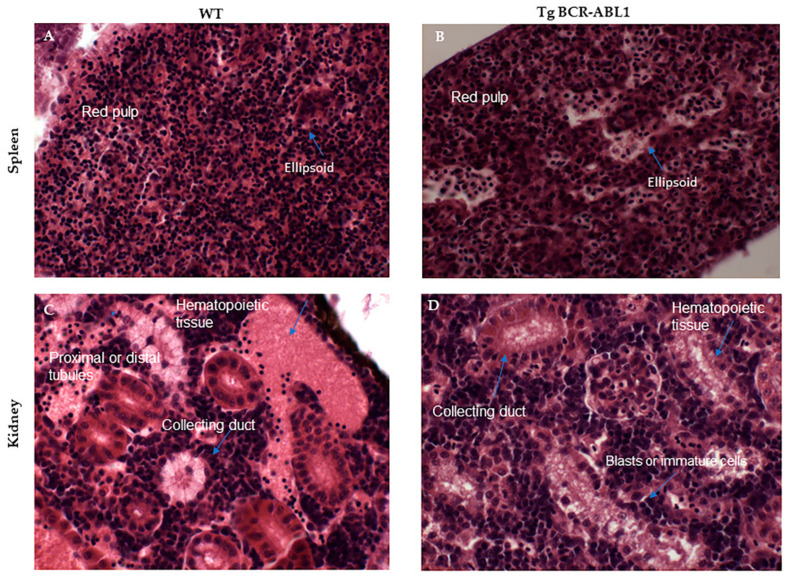
Hematoxylin-eosin staining on paraffin embedded longitudinal section (8 um) in wild type (WT) and tg BCR-ABL1 fish. Spleen (**A**,**B**) and kidney (**C**,**D**) are reported. Different histological and morphological structures are indicated blue arrows. (Magnification 40×) Bar 30 um.

**Table 1 cells-10-00445-t001:** List of primers used for Real Time PCR quantification.

Oligo	Sequence	Purpose
for1 BA cDNA	AGACTGTCCACAGCATTCCG	tg screening
rev1 BA cDNA	GCAACGAAAAGGTTGGGGTC	tg screening
for2 BA cDNA	GATGCTGACCAACTCGTGTG	tg screening
rev2 BA cDNA	GACCCGGAGCTTTTCACCTT	tg screening
for *scl*	CCGCTCGCCACTATTAACAG	real-time
rev *scl*	GTTCGTGAAAATCCGTCGC	real-time
for *lmo2*	ACTACAAACTCGGCAGAAAGC	real-time
rev *lmo2*	CACGCATGGTCATTTCAAAGG	real-time
for *gata1*	TGAATGTGTGAATTGTGGTG	real-time
rev *gata1*	ATTGCGTCTCCATAGTGTTG	real-time
for *pu.1*	CCATTAGAGGTGTCCGATGAG	real-time
rev *pu.1*	ACCAGATGCTGTCCTTCATG	real-time
for *runx1*	CCCCGCCCACAGCCAGATTC	real-time
rev *runx1*	GACGGGCGTGGGGGTGTAGGT	real-time
for *c-myb*	GGAAAGTGGAGCAAAGAAGGTTA	real-time
rev *c-myb*	TCGTGTAGTGTCTCTGGATAG	real-time
for *mpx*	TGATGTTTGGTTAGGAGGTG	real-time
rev *mpx*	GAGCTGTTTTCTGTTTGGTG	real-time

**Table 2 cells-10-00445-t002:** Different cell types in zebrafish peripheral blood. Myeloid cells include eosinophil and neutrophil granulocytes eosinophilic. The measurement is expressed in percentage. The counts were performed on 500 cells/field and 15 wild type and 15 tg BCR-ABL1 fish were analyzed.

Classification	Mature Red Cells	Myeloid Cells	Macrophages	Blasts Aggregates	Trombocytes
WT	57.6 ± 0.99	1.7 ± 0.98	2.5 ± 0.96	1.5 ± 0.07	0.13 ± 0.05
Tg BCR-ABL1	28.4 ± 0.97	10.4 ± 1.5	23.9 ± 1.6	15.2 ± 0.98	1.7 ± 1.3
*p* values	*p < 0.0001*	*p < 0.0001*	*p < 0.0001*	*p < 0.0001*	*p < 0.0001*

## Data Availability

Data are available by Corresponding author upon request.

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
