# Peer review of "Development of BCR-ABL1 Transgenic Zebrafish Model Reproducing Chronic Myeloid Leukemia (CML) Like-Disease and Providing a New Insight into CML Mechanisms"

_cells, 2021, doi:10.3390/cells10020445_

Round 1
Reviewer 1 Report
The authors describe the generation of a new transgenic zebrafish model resembling the features of the chronic phase of CML.
Major
Figure 1: Quality of the images is poor, (1C and D) please use higher resolution images. In particular Fig 1D lack any reference for the reader, maybe a transmission image (phase contrast?) should be superimposed to help the reader to “navigate” the image and provide e reference for the structures that can only be guessed.
Although the results are quite impressive I notice that there is no direct proof of the presence of expressed BCR/ABL protein. It is expected that tyrosine phosphorylation levels should also be increased in the model. This should be detectable by western blot analysis in zebrafish protein lysate (either direct analysis of cell lysate or following immunoprecipitation to increase the s/n ratio) or in situ by IF using anti ABL or antiPY antibodies. Absence of this type of data should be at least commented and reasons explained to the readers.
Beside figure 1 describing two positive transgenes there is no reported data on BCR/ABL mRNA expression or other form of quantification in whole embryo or blood. It would be interesting to associate , for example in figure 2, a corresponding quantification of BCR/ABL expression along with the proliferation and apoptotic markers.
The effect of BCR/ABL inhibitors should be tested to fully validate the model as they are expected at least partially revert the phenotype. This should represent a key information in order to validate the model for the use proposed (drug screening) and better compare the model with the one already present in the literature.
Minor
Data on long term effect in the transgenic animals (do the animals develop a blast phase?) is lacking
Author Response
We thank the reviewer for the deep analysis of the manuscript. We agree that most of the revisions suggested by the reviewer are appropriate and we tried to improve the manuscript according to those suggestions.
Major
Figure 1: Quality of the images is poor, (1C and D) please use higher resolution images. In particular Fig 1D lack any reference for the reader, maybe a transmission image (phase contrast?) should be superimposed to help the reader to “navigate” the image and provide e reference for the structures that can only be guessed.
Although the results are quite impressive I notice that there is no direct proof of the presence of expressed BCR/ABL protein. It is expected that tyrosine phosphorylation levels should also be increased in the model. This should be detectable by western blot analysis in zebrafish protein lysate (either direct analysis of cell lysate or following immunoprecipitation to increase the s/n ratio) or in situ by IF using anti ABL or antiPY antibodies. Absence of this type of data should be at least commented and reasons explained to the readers.
Thank you for this point. In order to improve the quality of Figure 1 (line 281) and to better show the protein expression in CFP positive cells, we repeated the experiments and we performed the heat treatment to activate the transgene and induce the protein expression. We also organized the Figure 1 in panel A and panel B. In panel A, we included a bright light image where we show the region corresponding to inner cell mass at 22 hpf. It is possible to observe the fluorescent CFP positive circulating cells both at 22 and 24 hpf that proved the expression of BCR-ABL1 protein. The experiment was done twice and we analyzed 15 embryos for each experiment at two different stages. Please, see Figure 1 Panel A and lines 275-279. In Figure 1 Panel B (line 285), the monitoring of BCR-ABL1 transcript level is reported. BCR-ABL1 transcript was quantified by digital PCR on embryos pools at 24, 48 and 72 hpf. Digital PCR graph represents the emission of fluorescence in the micro-reactions. Every pool was quantified twice on 2 different chips, each one divided in 20.000 micro reaction. Yellow dots represent negative micro-reaction (no emission), while blue dots represent positive reactions (emission in FAM). Each positive reaction contains one or two molecules of BCR-ABL1 transcript. The presence of blue dots in tg BCR-ABL1 embryos at 24, 48 and 72 hpf confirms the expression of the gene in the transgenic samples.
Beside figure 1 describing two positive transgenes there is no reported data on BCR/ABL mRNA expression or other form of quantification in whole embryo or blood. It would be interesting to associate , for example in figure 2, a corresponding quantification of BCR/ABL expression along with the proliferation and apoptotic markers.
We really thank the Reviewer for this suggestion. In order pair the quantification of BCR-ABL1 expression along with proliferation and apoptotic markers, we repeated the quantification of cells by Bromouridine staining at different developmental stages (24, 48 and 72 hpf). Please, see Figure 2 panel B (line 332) and the paragraph 3.2 line (300-302). In the original Figure S3, we showed images of embryos at 24 and 48 hpf stained with acridine orange. Fluorescence was quantified using Z-Mapper software at two different stage respectively 24 and 48 hpf (figure 2 panel B line 332). The results strongly suggest a reduced apoptosis in TG BCR-ABL1 fish. The quantification of BCR-ABL1 transcript was replicated both on tg BCR-ABL1 embryos and on WT fish at 24, 48 and 72 pfu. The details concerning the BCR-ABL1 transcript are reported in the previous point.
The effect of BCR/ABL inhibitors should be tested to fully validate the model as they are expected at least partially revert the phenotype. This should represent a key information in order to validate the model for the use proposed (drug screening) and better compare the model with the one already present in the literature.
We appreciate this suggestion and it is our intention to validate the model using a drug screening approach. Recent studies demonstrated that zebrafish shares 82% of disease-associated targets and several drug metabolism pathways with humans. We will explore if the pharmacological mechanisms in the established transgenic line mimics the CML patients’ one administering drugs used for CML patients’ treatment (imatinib, nilotinib, dasatinib and bosutinib). Our experimental plans as first was to obtain a stable transgenic line and be sure that the transgene was inherited and then, as shown in this paper, characterized the mayor phenotype of adult fish (paragraph 3.5 and 3.6). We recently asked to Italian Ministry of Health the approval of a project to test different molecules with pharmacological effect. By the Italian low, the animal models need to be declared as disease models before the administration of drug molecules and it is contingent upon the publication of the model. As soon as we will receive the permission, the drug screening will start.
Minor
Data on long term effect in the transgenic animals (do the animals develop a blast phase?) is lacking.
In humans, the natural progression of untreated CML is bi or tri-phasic, with initial chronic phase followed by acute phase, blast phase or both. Chronic phase is characterized by leukocytosis in both peripheral blood and bone marrow and a preponderance of granulocytes in various degrees of maturation. As the disease progresses, patients enter in acute phase followed by blast phase, during which hematopoietic differentiation arrests, allowing immature blasts to accumulate in the bone marrow. A level of 10-19 % of blasts marks the transition from chronic phase to blast phase. Adults of the transgenic stable line (fish 22-24 months old) present an increase of blast cells in a rate of 10% in peripheral blood. Moreover, the animals are smaller in compare to wild types at the same age, swim slower and present skeletal alterations (data not shown). We are going to better characterize this phenotype after the recognition as disease model. We added these comments in the discussion. Please, see line 582.
Finally, we include in the revised paper, also if was not requested, a new version of figure number 7 (Quantification of different cell types in peripheral blood) (line470-475)
We hope that our explanations satisfied the requests and really thank the Reviewer for his suggestions which have helped us in improving the quality of the manuscript.
Reviewer 2 Report
The authors present a model of CML disease in zebrafish by generating a transgenic line overexpressing the BRC-ABL1 oncogene. In March 2020 a similar zebrafish model was published by Xu et al and few minor differences are present between these two models and the analysis performed. Both model rely on lrval heat shock to activate the hBRC-ABL1 transgene in a ubiquitous manner. In the present manuscript the authors use a UAS/GAL4 system to amplify the expression but use it in combination with a HSP iducible Gal4 trangenic line.
Main point: Did they try to express BRC-ABL1 with other either constitutive or tissue specific promoters? For instance the use of a hematopoietic specific promoter would have been very informative and would have improved the animal model by mimicking more closely the human disease condition.
Specific points: Fig 1A the schematic is of low quality and the arrow between the two vectors is partly covered. Fig 1C Impossible to read the graphs. What is the point of this analysis? Fig 1D What is the region imaged? Very low quality pictures should be better oriented and annotated.
Author Response
We thank the reviewer for the deep analysis of the manuscript. We agree that most of the revisions suggested by the reviewer are appropriate and we tried to improve the manuscript according to those suggestions.
The authors present a model of CML disease in zebrafish by generating a transgenic line overexpressing the BRC-ABL1 oncogene. In March 2020 a similar zebrafish model was published by Xu et al and few minor differences are present between these two models and the analysis performed. Both model rely on lrval heat shock to activate the hBRC-ABL1 transgene in a ubiquitous manner. In the present manuscript the authors use a UAS/GAL4 system to amplify the expression but use it in combination with a HSP iducible Gal4 trangenic line.
Main point: Did they try to express BRC-ABL1 with other either constitutive or tissue specific promoters? For instance the use of a hematopoietic specific promoter would have been very informative and would have improved the animal model by mimicking more closely the human disease condition.
It is known that spi1 (pu.1) gene plays a key role in the development of monocytes, B lymphocytes, and granulocytes and it is expressed in a range of hematopoietic cells particularly myeloid expression is pronounced. Expression of spi1 marks the early compartment of zebrafish myelopoiesis, it starts at early stages of development and after 22 hpf, spi1 expression is observed in the posterior intermediate cell mass (ICM), and ultimately in the adult kidney, the site of zebrafish hematopoiesis. Given this expression pattern, the stable transgenic line tg(spi1:EGFP) is a useful tool for studies in zebrafish myeloid cells in a living zebrafish embryo (A.C Ward Blood 2003). We got the stable transgenic line tg spi1:EGFP; recently we crossed this transgenic line with tg BCR-ABL1 creating a double transgenic and inducible line. The definition and characterization of the tg fish line described in the present manuscript is mandatory for the further experimental plans. The double and stable transgenic line will allow us to study and characterize directly the behavior and the involvement of myeloid cells line in a model mimicking the human CML disease.
Specific points: Fig 1A the schematic is of low quality and the arrow between the two vectors is partly covered. Fig 1C Impossible to read the graphs. What is the point of this analysis? Fig 1D What is the region imaged? Very low quality pictures should be better oriented and annotated.
Thank you for this point. In order to improve the quality of Figure 1 (line 281) and to better show the protein expression in CFP positive cells, we repeated the experiments and we performed the heat treatment to activate the transgene and induce the protein expression. We also organized the Figure 1 in panel A and panel B. In panel A, we included a bright light image where we show the region corresponding to inner cell mass at 22 hpf. It is possible to observe the fluorescent CFP positive circulating cells both at 22 and 24 hpf that proved the expression of BCR-ABL1 protein. The experiment was done twice and we analyzed 15 embryos for each experiment at two different stages. Please, see Figure 1 Panel A and lines 275-279. In Figure 1 Panel B (line 285), the monitoring of BCR-ABL1 transcript level is reported. BCR-ABL1 transcript was quantified by digital PCR on embryos pools at 24, 48 and 72 hpf. Digital PCR graph represents the emission of fluorescence in the micro-reactions. Every pool was quantified twice on 2 different chips, each one divided in 20.000 micro reaction. Yellow dots represent negative micro-reaction (no emission), while blue dots represent positive reactions (emission in FAM). Each positive reaction contains one or two molecules of BCR-ABL1 transcript. The presence of blue dots in tg BCR-ABL1 embryos at 24, 48 and 72 hpf confirms the expression of the gene in the transgenic samples.
Finally, we include in the revised version an update of figure 2. In order to have more evidences about the quantification of BCR/ABL expression along with proliferation and apoptotic markers we repeated quantification of cells by Bromouridine staining at different developmental stages (24, 48 and 72 hpf ) as shown in figure 2 panel B (line 332) and described in the paragraph 3.2 line (300-302). In the original Figure S3 we showed images of embryos at 24 and 48 hpf stained with acridine orange. Fluorescence was quantified using Z-Mapper software at two different stage respectively 24 and 48 hpf (figure 2 panel B line 332). The results strongly suggest a reduced apoptosis in TG BCR-ABL1 fish.
A revised version of Figure 7 (Quantification of different cell types in peripheral blood) (line470-475) was added.
We hope that our explanations satisfied the request of the reviewer and really thank for the suggestions which helped us in improving the quality of the manuscript.
Round 2
Reviewer 1 Report
The authors have satisfactorily responded to the remarks. I only want to underline that in figure 1 panel B I can not see the blue dots. Whether is a problem with my viewer or a low resolution image from the original figure is a matter of evaluation for authors and editor.
Author Response
Reviewer 1
We thank the reviewer for the new analysis of the manuscript. We think that the minor revision suggested by the reviewer is appropriate and we tried to improve the Figure 1 panel B according to that suggestions.
Minor
The authors have satisfactorily responded to the remarks. I only want to underline that in figure 1 panel B I can not see the blue dots. Whether is a problem with my viewer or a low resolution image from the original figure is a matter of evaluation for authors and editor.
Thanks for this minor point. We improved the quality of Figure 1 panel B and provided a new figure of High Resolution quality where it is possible to appreciate the blue dots (line 284), marked with red arrows. The blue dots represent positive reactions (emission in FAM). Each positive reaction contains one or two molecules of BCR-ABL1 transcript. The presence of blue dots in tg BCR-ABL1 embryos at 24, 48 and 72 hpf confirms the expression of the gene in the transgenic samples while yellow dots represent negative micro-reaction (no emission).
We hope that explanation satisfied the minor request and really thank the reviewer for his suggestion.

Reviewer 2 Report
The authors have made some efforts to address my previous points.
Even if not all the main points raised were addressed, I'm happy to accept the manuscript in the present form.
Author Response
Reviewer 2
We thank the reviewer for the analysis of resubmitted manuscript.
The authors have made some efforts to address my previous points. Even if not all the main points raised were addressed, I'm happy to accept the manuscript in the present form.
Previous suggestions and revisions allowed us in improving the quality of manuscript.
